# Exploring Dietary Supplement Utilization Patterns Among African American Survivors of Prostate and Breast Cancer: A Cross-Sectional Analysis

**DOI:** 10.3390/nu17233724

**Published:** 2025-11-27

**Authors:** Carlene A. Kranjac, Patricia Sheean, Anjishnu Banerjee, Bi Qing Teng, Kathleen O’Connell, Margaret Tovar, Estefania Alonso, Zoe Snider, Melinda Stolley

**Affiliations:** 1Department of Medicine, Medical College of Wisconsin, Milwaukee, WI 53226, USAmstolley@mcw.edu (M.S.); 2Parkinson School of Health Sciences and Public Health, Loyola University Chicago, Maywood, IL 60153, USA; psheean1@luc.edu; 3Data Science Institute, Medical College of Wisconsin, Milwaukee, WI 53226, USA; 4Department of Psychology, University of Pittsburgh, Pittsburgh, PA 15260, USA

**Keywords:** dietary supplements, cancer survivorship, prostate cancer, breast cancer, complementary and alternative medicine

## Abstract

**Background/Objectives:** Cancer survivors are a growing population in the United States, with projections of 22.5 million by 2030. Cancers of the prostate (PC) and breast (BC) are among the most prevalent. Despite a high burden of disease, African American survivors are underrepresented in health behavior research. Leveraging two large databases, this study uniquely characterizes dietary supplement (DS) use among African American cancer survivors to explore potential intervention points. **Methods:** Characteristics from 376 African American cancer survivors (130 PC; 246 BC) in lifestyle intervention trials were examined. DS use was self-reported and categorized by type. A logistic regression model examined associations between use and survivor characteristics. **Results:** Overall, 215 (63.80%) survivors with baseline medication log data (N = 337) reported using at least one DS, with a higher prevalence among BC survivors (67.44%) than PC survivors (57.38%). Vitamin D ± calcium combinations, multivitamins, omega-3 fatty acids, calcium, Vitamin B12, and Vitamin C were the most frequently reported. Total comorbidities (mean = 2.38, SD = 1.66) significantly predicted increased DS use among BC and PC survivors. Educational attainment (≤12th grade vs. graduate/professional education) and diet quality (high vs. low) were significantly associated with lower odds of DS use for PC survivors. Only diet quality (moderate vs. low) was significantly associated with higher odds of DS use in BC survivors. **Conclusions:** DS use is common among African American PC and BC survivors participating in lifestyle interventions. These findings underscore the need for evidence-based guidelines regarding DS use among cancer survivors and the need to include diverse populations.

## 1. Introduction

In the United States (U.S.), cancer rates are continuing to rise, with over two million diagnoses in 2024 [1]. With increased incidence, along with new treatment modalities and technology, cancer survivors are a growing community projected to increase from 18.6 million in 2025 to 22.5 million by 2030 [1,2]. Breast cancer (BC) is the most commonly diagnosed cancer in women with an estimated 316,000 new cases of invasive BC in 2025, whereas prostate cancer (PC) is the most commonly diagnosed cancer in men with an estimated 313,780 new cases in 2025 [3]. Owing to a complex interplay of biologic, environmental, and policy factors, both cancers have significant disparities with African Americans developing more aggressive disease at younger ages compared to their White counterparts [4]. Recent estimates for PC incidence and mortality over the past decade were 1.67-fold and 2.05-fold higher, respectively, for Non-Hispanic Black men compared to Non-Hispanic White men [5]. Regarding BC, incidence was lower for Non-Hispanic Black women (0.95-fold) compared to Non-Hispanic White women, but mortality was higher (1.38-fold) [5]. While disparities are also evident for survival, the majority of African American individuals diagnosed with BC or PC will survive at least 5 years [5]. Thus, efforts to enhance survivorship are of value. Studies have demonstrated the significant impact that lifestyle, including diet and exercise, can have on PC and BC outcomes; however, many studies have not included racially representative samples [6,7].

To address this gap, the Men Moving Forward (MMF) and Moving Forward (MF) randomized trials were conducted to examine the efficacy of community and group-based, supervised lifestyle programs on promoting changes in body mass index (BMI), body composition, and patient-reported outcomes of African American PC and BC survivors, respectively [8]. The interventions reflected recommendations of the American Cancer Society (ACS)’s physical activity and nutrition guidelines for cancer survivors, recommending plant-based dietary patterns, regular physical activity, and resistance exercise training [8,9]. Dietary supplements (DSs) were not addressed, as the ACS guidelines do not support the use of DSs for cancer prevention, and do not provide definitive guidance for their use in cancer survivors [10].

DSs, as defined by the ACS, include “vitamins and minerals, herbs and other botanicals, amino acids, enzymes, and more” [10]. DS use is common, with more than 50% of adults in the U.S. reporting use of at least one DS and 33% reporting use of multivitamin/multimineral supplements [11]. Moreover, between 64 and 81% of all cancer survivors and between 14 and 32% of individuals newly diagnosed with cancer will begin taking DSs to support their immune health, reduce stress, and improve health outcomes [11]. The limited existing literature suggests that DS use is not uncommon among cancer survivors. However, most studies to date have included samples comprising predominantly Non-Hispanic White cancer survivor populations and report inconclusive evidence for their potential health impacts [11,12]. For example, some studies suggest that the antioxidant properties of Vitamin C supplementation may negatively influence the metabolism and action of chemotherapy medications [13]. However, other studies have reported that when used as a high-dose form, it may aid in reducing metastatic spread in individuals with a terminal diagnosis [13]. Additionally, recent evidence suggests that Vitamin D supplementation after a cancer diagnosis could lead to improved survival; however, randomized controlled trials fail to establish conclusive evidence [14]. Studies looking at the impact of calcium emphasize the positive impact on bone health and some even suggest decreased mortality, yet this has not been largely reproducible [15,16]. Furthermore, preclinical studies on Vitamin B12 have demonstrated increased antitumor activity, but similar findings have yet to be established in clinical trials [17]. Lastly, use of multivitamins as DSs as has not provided a clinically significant effect in cancer recurrence or survival but may be hard to classify given the variety of multivitamins that exist [18]. Overall, these findings highlight the inconclusive nature of existing evidence regarding DSs and their effects on cancer-related outcomes, highlighting the need to better understand the patterns and impacts of DS use across populations.

To provide needed cultural context and expand survivorship representation, we describe DS use and correlates of use among African American survivors of early-stage BC and PC engaged in lifestyle intervention trials. Given the dearth of research to date, the results of this project will provide novel findings as one of the first studies to investigate DS use within this population. This foundational information can be used to support future research on the safety and clinical implications of DS use in cancer survivorship, ultimately supporting the development of evidence-based guidelines.

## 2. Materials and Methods

### 2.1. Study Design, Eligibility, and Recruitment

This cross-sectional study optimized baseline data from 376 study participants engaged in the Moving Forward (N = 246) and Men Moving Forward (N = 130) lifestyle interventions. Men Moving Forward (MMF) is a randomized-controlled lifestyle intervention study for African American PC survivors (2020–2025) examining the efficacy of a community-based intervention promoting adherence to the ACS Nutrition and Physical Activity Guidelines. To be eligible, men must be ≥ 18 years, have been diagnosed with PC, have completed treatment at least 6 months before starting the intervention or be on active surveillance, and have a BMI of ≥ 23 kg/m^2^. Moving Forward (MF) is a completed randomized controlled lifestyle intervention study reporting the efficacy of a guided weight loss program for African American BC survivors (2010–2015). To be eligible, women had to be ≥ 18 years of age, diagnosed with BC, completed treatment at least 6 months before starting the intervention, and have a BMI ≥ 25 kg/m^2^. Additional criteria for both studies included not meeting the ACS guidelines for daily fruit and vegetable consumption or physical activity. Although these studies were conducted at different timepoints, both had comparable study designs, intervention structures, and data collection protocols.

Recruitment efforts included several strategies. The primary strategy for both studies relied on patient lists from the cancer registries of urban academic cancer centers. Patients diagnosed with non-metastatic PC or BC received mailings describing the study, followed by phone calls to assess interest and determine eligibility. Additional efforts included outreach to community partners through our community advisory boards, including community-based organizations, faith-based institutions, community centers, community leaders and community clinics. Once a participant related interest, physician approval regarding safety to participate in the lifestyle intervention trials was obtained. All participants provided written informed consent prior to participation. These studies were approved by the Institutional Review Board at the Medical College of Wisconsin.

### 2.2. Measures

DS use, demographic data, and health histories were collected via interview by trained research staff. DS use was asked about as part of the medication self-report log at a baseline visit, prior to randomization. Participants were asked to report all medications they were currently taking, including prescription and over-the-counter medication, vitamins, minerals, supplements, or non-drug therapies. For each reported DS, participants provided the product name and frequency of use. Participants provided this information from memory, with lists of their medications, with pictures of the label, or by bringing in the DS to the baseline appointment. DS were subsequently categorized by type (see Appendix A) and entered into the online research study database; all entries were verified by two research staff.

Demographic data included self-reported age, education, marital status, and income, and clinical data included co-morbidities. BMI, also part of clinical data, was collected during a physical assessment, and calculated as weight (kg)/height (m^2^). Height was measured to the nearest 0.1 cm using a portable stadiometer, and weight was measured to the nearest 0.1 kg using a digital scale. If there was more than a 0.2 kg weight or 0.5 cm height discrepancy, a third measurement was taken. BMI was calculated based on the mean of the two most similar height and weight measurements.

Lifestyle data included self-reported physical activity and dietary intake. Physical activity was measured by the Godin Leisure Activity Scale. The Godin leisure time exercise survey measures the number of 15 min exercise periods over 7 days, and evidence has supported its validity and utility in classifying physical activity [19]. Godin scores can be categorized into “insufficiently active (score < 14)”, “moderately active (score 14–23)”, and “active (score ≥ 24)”. Usual dietary intake was captured using a food frequency questionnaire administered via interview, from which a Healthy Eating Index (HEI) score was calculated. The Block 2005 Food Frequency Questionnaire was used with BC survivors, while the Vioscreen Food Frequency Questionnaire was used with PC survivors. The HEI is a measure used to assess diet quality and is updated every 5 years. This index measures intake of fruits, vegetables, greens and beans, whole grains, dairy, protein, added sugars, saturated fats, refined grains, and sodium to assess diet quality in relation to the Dietary Guidelines for Americans (DGA) [20]. Baseline dietary intake was classified by the 2010 HEI into “low diet quality (score < 51)”, “moderate diet quality (score 51–80)”, and “high diet quality (score > 80)”.

### 2.3. Statistical Analyses

Descriptive statistics were used to summarize baseline characteristics across the two cancer survivor groups in the full cohort of participants (N = 376). Thirty-nine participants did not report DS use on their baseline visit medication logs (31 BC and 8 PC) and thus were categorized as “Unknown” and not included in the DS prevalence and intake analyses. Variables were summarized by the mean and standard deviation for continuous variables, and frequency and percentage for categorical variables.

A logistic regression model was used to examine which baseline characteristics were associated with DS use in each study separately. The model included the following baseline variables: age, education, income, BMI, Godin categories, HEI, and total number of comorbidities. Observations with missing DS data, BMI, and total comorbidities were excluded from the fitting model (N = 31 BC and N = 13 PC) as missing data do not allow for meaningful estimates. For categorical variables with missing values, a “Missing” category was created to retain these observations in the model. Categories with small counts were grouped (i.e., eighth grade or less and some high school). A stepwise backward selection method based on the Akaike Information Criterion (AIC) was used to determine which covariates were included in the final model. In the final model for BC survivors (Moving Forward), there were 215 observations. The final model included education, HEI score categories, and total comorbidities. In the final model for PC survivors (Men Moving Forward), there were 117 observations. The final model included age, education, HEI score categories, and total comorbidities. Odds ratios, 95% confidence intervals, and *p*-values were calculated for each of the baseline variables included in the final model. We used a significance level of 0.05 to determine whether each variable was associated with higher or lower odds of the outcome. All analyses were performed using R version 4.0.3.

## 3. Results

### 3.1. Study Population Characteristics

Characteristics of the BC survivors and PC survivors participating in MF and MMF are described in Table 1 (N = 376). Overall, PC survivors were older and had a lower BMI compared to BC survivors. Both groups reported a similar number of co-morbidities, with the most common being hypertension. Sociodemographic and lifestyle characteristics also differed between groups. A larger proportion of PC survivors had some college education and fewer had a combined family income less than $20,000 compared to BC survivors. Moreover, a greater number of BC survivors were categorized as insufficiently active based on the Godin score compared to PC survivors. Lastly, both groups demonstrated largely moderate diet quality according to HEI scores.

### 3.2. Prevalence of Dietary Supplement Use

Of all study participants with available DS use data (N = 337), the majority (N = 215) reported using at least 1 DS (Table 2). A subset of individuals (N = 39) were excluded due to missing data—31 BC and 8 PC—these data are intentionally included and reported as “Unknown” in Table 2. Overall, baseline DS use was higher among BC survivors compared to PC survivors and a large subset of participants (N = 122) reported not using any DSs.

Across both studies, DS use was common with 20.77% (N = 70) of participants using at least one DS and many reporting the use of multiple DSs (Table 2). BC survivors tended to use DSs at similar, but slightly higher, percentages than PC survivors. Both groups included participants taking one, two, or three DSs with BC survivors having higher DS use in all categories (Table 2, Figure 1).

### 3.3. Types of Dietary Supplements Used

Both intervention study populations who reported DS use (N = 337) reported using the same six DSs most often. These included Vitamin D or a combination of Vitamin D and calcium, multivitamins with or without minerals, calcium, omega-3 fatty acids, Vitamin B12, and Vitamin C (Table 2) (Figure 2).

### 3.4. Predictors of Dietary Supplement Use

#### 3.4.1. Prostate Cancer Survivors

The final model for PC survivors included age, education level, HEI score, and total comorbidities, representing 117 PC survivors (Table 3a). Age was not statistically significant among PC survivors; however, regarding education, those with some high school or 8th grade or less had lower odds of DS use compared to those with a graduate or professional degree. Regarding diet quality, PC survivors reporting high diet quality (score > 80) had lower odds of using DSs compared to those who reported low diet quality (score < 51). Finally, for every additional comorbidity that a PC survivor had, they had higher odds of using DSs.

#### 3.4.2. Breast Cancer Survivors

The final model for BC survivors included education level, HEI score, and total comorbidities, representing 215 BC survivors (Table 3b). No statistically significant association was observed between education level and DS use among BC survivors. However, BC survivors who reported moderate diet quality (score 51–80) had increased odds of DS use compared to those with low diet quality (score < 51). Finally, BC survivors had higher odds of using DS with every additional comorbidity.

## 4. Discussion

This study is one of the first to focus on DS use among African American BC and PC survivors, a population significantly under-represented in DS and cancer survivorship research. In this cross-sectional analysis, most study participants used at least one DS, and the six most-used DSs for both groups included Vitamin D ± calcium combinations, multivitamins, omega-3 fatty acids, calcium, Vitamin B12, and Vitamin C. DS use varied with demographic, lifestyle, and clinical characteristics. Among PC survivors, those with less education and higher diet quality were less likely to report DS use. For BC survivors, those with moderate diet quality (when compared to those with low diet quality) were more likely to use DS. An increase in co-morbidities for both PC and BC survivors significantly increased the odds of using DSs. Additionally, BC survivors utilized more DSs than PC survivors, suggesting that gender may have an influence as well.

Overall, DS use in our study population (63.8%) is similar to but slightly lower than previous reports among all U.S. cancer survivors (70.4%) [21]. Higher DS use among our study’s BC survivors compared to PC survivors (67.44% and 57.38%, respectively) is also consistent with previous studies showing that BC survivors report the highest DS use, while PC survivors report the lowest [11,12]. In relation to individuals in the general population, 2023 National Health and Nutrition Examination Survey (NHANES) data showed that 39.3% of Non-Hispanic Black U.S. adult men and 56.2% of Non-Hispanic Black U.S. adult women without cancer use DSs [22]. This is much lower than the prevalence of DSs reported among our PC survivors (57.4%) and BC survivors (67.4%). Furthermore, while our results align with prior research showing that DS use varies by demographic, lifestyle, and clinical factors, our findings also differ from prior studies conducted with predominantly Non-Hispanic White samples. For example, in studies with survivors of mixed cancer types, DS use is greater among Non-Hispanic Whites, those reporting female gender, and individuals with higher education and income [11,23,24,25]. However, our findings show that while PC survivors with less education use fewer DSs, there were no significant associations in BC survivors. Additionally, consistent with these prior studies, female gender was associated with greater DS use in our study.

The existing literature also suggests that lower BMI, increased physical activity, and higher diet quality are associated with DS use among cancer survivors [26,27]. Our results both support and challenge these findings. Whereas BC survivors with moderate diet quality had higher DS use, PC survivors with high diet quality had lower use. It is possible that PC survivors with low diet quality used DSs to make up for nutrition deficits, and those with higher diet quality did not feel like they needed nutritional supplementation. A further distinction of our study results is the strong association between comorbidities and increased DS use across both samples. Prior work supports higher DS use among those with cancer, high cholesterol, arthritis, and eye disease [28], and lower DS use in persons with diabetes mellitus [28]. Thus, our findings on the clinical characteristics associated with DS are important as they can enable providers to deliver more comprehensive, high-quality care through patient-centered discussions exploring patient beliefs about DSs, addressing misinformation and improving communication to support safe and shared decision-making for DS use in cancer survivors.

To better inform clinical discussions regarding DS use, it is important to know which DSs are most often consumed by cancer survivors. Among our cancer cohorts, these included: multivitamins ± minerals, Vitamin D ± calcium, omega-3 fatty acids, Vitamin B12, and Vitamin C. These observations are comparable to previous reports. Specifically, fish oils, calcium ± Vitamin D, multivitamins ± minerals, and Vitamin D were the most prevalent DSs reported in a cross-sectional survey (N = 1049) of breast, prostate, and colorectal cancer survivors [12]. Although this survivor behavior is very common, a recent investigation highlights no tangible health benefits of taking even a multivitamin/mineral within the general population [29]. Interestingly, a 2020 systematic review and meta-analysis of observational and randomized controlled trials examining antioxidant use (Vitamin C, Vitamin E, and carotenes) found no overall association between these DSs and a decreased risk of all-cause mortality or cancer recurrence among a heterogenous group of cancer survivors [15]. However, in BC-specific studies, post-cancer diagnosis of the use of Vitamin C, Vitamin D, and Vitamin E was associated with a reduced risk of premature mortality, while multivitamins, Vitamin C, and Vitamin E were associated with decreased BC recurrence [15]. In contrast, an earlier study of antioxidant use, including Vitamins C and E, in post-menopausal women with BC reported worsened prognosis when these were used concurrently with chemotherapy or radiation [30]. In the PC setting, antioxidant and DS use, such as Vitamins C, D, and E, among others, has not been found to affect prostate-specific antigen (PSA) levels, making their impact on PC risk unclear, and ultimately, producing inadequate results to recommend the use of DSs in PC [31].Despite establishing patterns of use and identifying common DSs used by cancer survivors, understanding the clinical implications of these behaviors remains challenging.

We speculate that there are several reasons why survivors consume DS and offer several reasons regarding the potential difficulties in drawing conclusions surrounding their purported risks and/or health benefits. First, the Dietary Supplements Health Education Act of 1994 (DSHEA) provides few regulations for DSs, including a label that claims the DS has not been reviewed by the U.S. Food and Drug Administration (FDA) [32]. While the DSHEA prohibits manufacturers from making disease treatment claims, it permits manufacturers to claim their product provides nutritional support [32]. Even with these regulations, consumers often perceive DSs as capable of preventing or treating specific diseases [33]. As such, labs can make speculative claims and exploit cancer survivors’ consumer awareness. Second, there is wide variability for the ingredients contained within DS. Investigations into the composition of DSs have revealed the concomitant ingestion of flavonoids, polyethylene glycols, heavy metals, and microorganisms, such as fungal isolates [34,35]. Third, in the absence of standardized regulation, it is difficult to decipher the true impact of DSs on cancer-related outcomes [10]. Taken together, these findings and speculations highlight the inconclusive nature of existing evidence regarding DSs and their effects on cancer-related outcomes. While the ACS currently has no recommendations for DS use in cancer survivors, and the existing literature is often inconclusive, establishing clinical guidelines and practices for physicians is crucial to promoting overall health and quality of life, and preventing unnecessary costs and potential risks from DS use.

There are several limitations to these analyses that merit attention and context. First, data were self-reported by participants interested in lifestyle behaviors and, thus, prone to recall or social desirability bias. Wanting to provide socially acceptable responses and appear to live healthier may have motivated individuals to report DS use differently. Alternatively, participants may have failed to report DSs for fear of retaliation or judgment (“You aren’t going to tell my doctor, right?”). Together, these could lead to over- or underreporting. Second, the implications of missing data are simply unknown. We are unsure why these 39 individuals failed to adequately report DS use. We hypothesize it was largely due to recall difficulty, skipped items during their baseline visit, or they were lost to follow-up. Third, the reasons why participants were taking DSs, whether this was recommended by their physicians or if they initiated DS use on their own, were not collected in either study. Lastly, these studies were conducted in different years, and societal attitudes to and trends towards using DSs have varied, impacting reported DS use in the general US population [36]. As such, this trend supports that our findings may be conservative estimates, drawing further attention to this important topic.

## 5. Conclusions

Understanding the prevalence and patterns of DS use among African American cancer survivors addresses a notable gap in the current literature. Although this secondary analysis has noted limitations (i.e., missing data and lifestyle-oriented populations), these study findings provide an important foundation for future, more inclusive investigations into the safety, efficacy, and clinical implications of DS use in cancer survivorship. Further, these data may prompt additional lines of inquiry which can drive the development of evidence-based guidelines, reduce misinformation, and empower healthcare providers to engage in more informed conversations with their patients.

## Figures and Tables

**Figure 1 nutrients-17-03724-f001:**
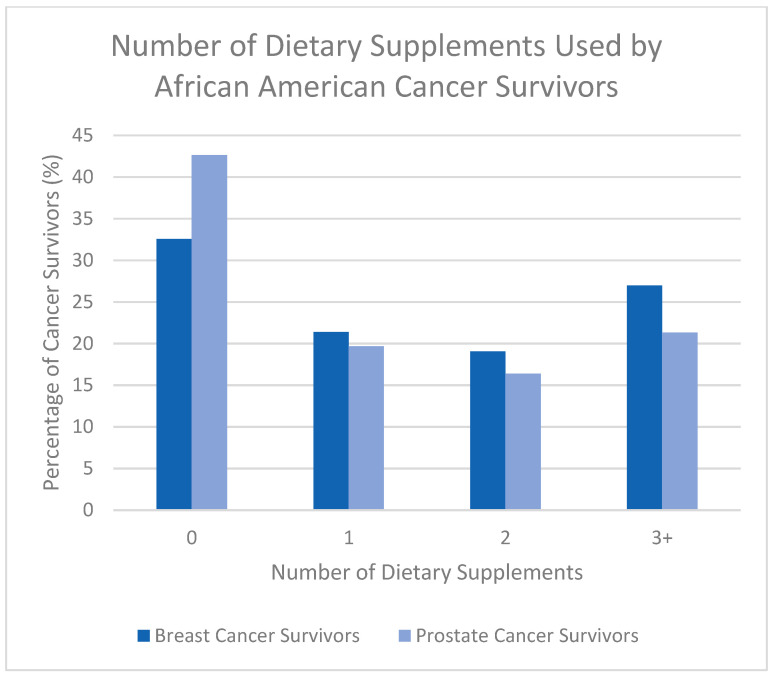
Number of dietary supplements used by African American breast cancer (BC) and prostate cancer (PC) survivors.

**Figure 2 nutrients-17-03724-f002:**
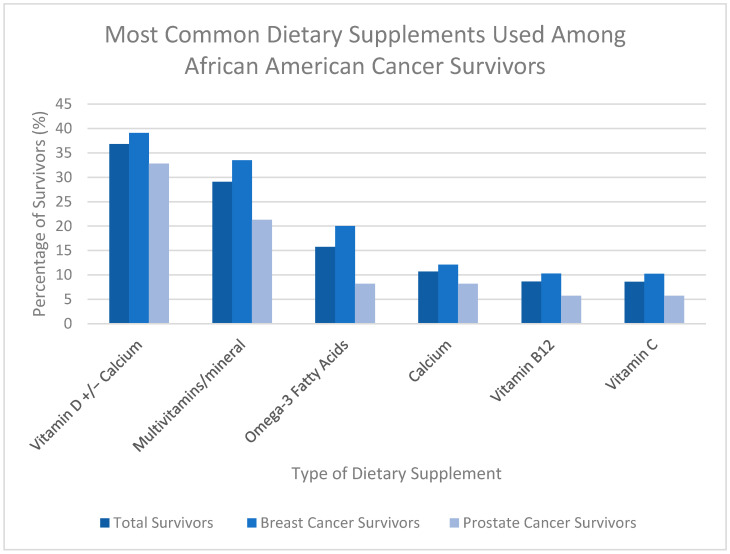
Percentage of African American breast cancer (BC) and prostate cancer (PC) survivors using the six most reported dietary supplements.

**Table 1 nutrients-17-03724-t001:** Baseline characteristics for the entire cohort of breast cancer (BC) and prostate cancer (PC) survivors enrolled in the lifestyle interventions Moving Forward (MF) and Men Moving Forward (MMF).

Characteristic	Overall, N = 376 ^1^	BC Survivors (MF), N = 246	PC Survivors (MMF), N = 130
Age (years)			
Mean (SD)	60.33 (9.96)	57.46 (10.09)	65.76 (7.06)
Unknown	0 (0.00%)	0 (0.00%)	0 (0.00%)
Highest grade or year of school completed			
Eighth grade or less	3 (0.81%)	0 (0.00%)	3 (2.38%)
Some high school	28 (7.53%)	12 (4.88%)	16 (12.70%)
High school graduate or GED	77 (20.70%)	47 (19.11%)	30 (23.81%)
Associates degree or 2-year certificate	49 (13.17%)	39 (15.85%)	10 (7.94%)
Some college	95 (25.54%)	54 (21.95%)	41 (32.54%)
College graduate	61 (16.40%)	47 (19.11%)	14 (11.11%)
Graduate or professional degree	58 (15.59%)	47 (19.11%)	11 (8.73%)
Other	1 (0.27%)	0 (0.00%)	1 (0.79%)
Unknown	4 (1.06%)	0 (0.00%)	4 (3.08%)
Combined family income for the last 12 months			
Less than $20,000	74 (19.89%)	58 (23.58%)	16 (12.70%)
$20,000–$39,999	77 (20.70%)	56 (22.76%)	21 (16.67%)
$40,000–$59,999	63 (16.94%)	48 (19.51%)	15 (11.90%)
$60,000–$79,999	49 (13.17%)	33 (13.41%)	16 (12.70%)
$80,000 or more	90 (24.19%)	50 (20.33%)	40 (31.75%)
Prefer not to answer/Refused	19 (5.11%)	1 (0.41%)	18 (14.29%)
Unknown	4 (1.06%)	0 (0.00%)	4 (3.08%)
Marital status			
Single, never married	86 (23.18%)	66 (26.94%)	20 (15.87%)
Married or living with a partner	162 (43.67%)	95 (38.78%)	67 (53.17%)
Separated	18 (4.85%)	13 (5.31%)	5 (3.97%)
Divorced	79 (21.29%)	52 (21.22%)	27 (21.43%)
Widowed	26 (7.01%)	19 (7.76%)	7 (5.56%)
Unknown	5 (1.33%)	1 (0.41%)	4 (3.08%)
Godin Score			
Mean (SD)	18.72 (18.64)	16.87 (18.78)	22.36 (17.88)
Unknown	5 (1.33%)	0 (0.00%)	5 (3.85%)
Godin Score			
<14: Insufficiently active (low benefits)	216 (58.22%)	175 (71.14%)	41 (32.80%)
14–23: Moderately active (some benefits)	54 (14.56%)	24 (9.76%)	30 (24.00%)
≥ 24: Active (substantial benefits)	101 (27.22%)	47 (19.11%)	54 (43.20%)
Unknown	5 (1.33%)	0 (0.00%)	5 (3.85%)
HEI score			
Mean (SD)	64.46 (11.30)	65.08 (11.09)	63.36 (11.64)
Unknown	30 (7.98%)	25 (10.16%)	5 (3.85%)
HEI score			
<51: Low diet quality	43 (12.43%)	23 (10.41%)	20 (16.00%)
51–80: Moderate diet quality	269 (77.75%)	174 (78.73%)	95 (76.00%)
>80: High diet quality	34 (9.83%)	24 (10.86%)	10 (8.00%)
Unknown	30 (7.98%)	25 (10.16%)	5 (3.85%)
BMI			
Mean (SD)	34.31 (6.43)	36.13 (6.25)	30.72 (5.15)
Unknown	6 (1.60%)	0 (0.00%)	6 (4.62%)
Total comorbidities			
Mean (SD)	2.38 (1.66)	2.34 (1.64)	2.48 (1.71)
Unknown	4 (1.06%)	0 (0.00%)	4 (3.08%)
High blood pressure	234 (62.90%)	145 (58.94%)	89 (70.63%)
Unknown	4 (1.06%)	0 (0.00%)	4 (3.08%)
Arthritis/back problems	155 (41.67%)	121 (49.19%)	34 (26.98%)
Unknown	4 (1.06%)	0 (0.00%)	4 (3.08%)
High cholesterol	153 (41.13%)	93 (37.80%)	60 (47.62%)
Unknown	4 (1.06%)	0 (0.00%)	4 (3.08%)
Diabetes	94 (25.27%)	56 (22.76%)	38 (30.16%)
Unknown	4 (1.06%)	0 (0.00%)	4 (3.08%)
Sleep apnea	68 (18.28%)	42 (17.07%)	26 (20.63%)
Unknown	4 (1.06%)	0 (0.00%)	4 (3.08%)

^1^ Mean (SD); n (%).

**Table 2 nutrients-17-03724-t002:** Prevalence, quantity, and most-used dietary supplements (DSs) at baseline for breast cancer (BC) and prostate cancer (PC) survivors.

Characteristic	Overall, N = 376 ^1,2^	BC Survivors (MF), N = 246 ^1,2^	PC Survivors (MMF), N = 130 ^1,2^
Cancer survivors taking at least 1 DS			
0	122 (36.20%)	70 (32.56%)	52 (42.62%)
1+	215 (63.80%)	145 (67.44%)	70 (57.38%)
Unknown	39 (10.4%)	31 (12.60%)	8 (6.15%)
Average number of DSs used			
Mean (SD)	1.68 (2.03)	1.87 (2.15)	1.36 (1.76)
Unknown	39 (10.4%)	31 (12.60%)	8 (6.15%)
Number of DSs used			
0	122 (36.20%)	70 (32.56%)	52 (42.62%)
1	70 (20.77%)	46 (21.40%)	24 (19.67%)
2	61 (18.10%)	41 (19.07%)	20 (16.39%)
3+	84 (24.93%)	58 (26.98%)	26 (21.31%)
Unknown	39 (10.4%)	31 (12.60%)	8 (6.15%)
Vitamin D or combination of Vitamin D and calcium use	124 (36.80%)	84 (39.07%)	40 (32.79%)
Unknown	39 (10.4%)	31 (12.60%)	8 (6.15%)
Multivitamin or multivitamin with multiminerals	98 (29.08%)	72 (33.49%)	26 (21.31%)
Unknown	39 (10.4%)	31 (12.60%)	8 (6.15%)
Calcium	36 (10.68%)	26 (12.09%)	10 (8.20%)
Unknown	39 (10.4%)	31 (12.60%)	8 (6.15%)
Omega-3 fatty acids	51 (15.73%)	43 (20.00%)	10 (8.20%)
Unknown	39 (10.4%)	31 (12.60%)	8 (6.15%)
Vitamin B12/Cobalamin	29 (8.63%)	22 (10.28%)	7 (5.74%)
Unknown	40 (10.64%)	32 (13.01%)	8 (6.15%)
Vitamin C	29 (8.61%)	22 (10.23%)	7 (5.74%)
Unknown	39 (10.4%)	31 (12.60%)	8 (6.15%)

^1^ This table includes participants with missing DS data. They are classified as “Unknown” (N = 39). ^2^ Mean (SD); n (%).

**Table 3 nutrients-17-03724-t003:** (**a**) Predictors of dietary supplement (DS) use among prostate cancer (PC) survivors shown as odds ratios (ORs) with 95% confidence intervals (CIs) and *p*-values from logistic regression analyses (N = 117) ^1^. (**b**) Predictors of dietary supplement (DS) use among breast cancer (BC) survivors shown as odds ratios (ORs) with 95% confidence intervals (CIs) and *p*-values from logistic regression analyses (N = 215) ^1^.

(**a**)			
**Characteristic**	**OR ^2^**	**95% CI ^2^**	***p*-Value ^3^**
Age	1.06	0.997, 1.13	0.067
Education			
Graduate or professional degree	—	—	
Eighth grade or less/some high school	0.12	0.02, 0.72	0.025
High school graduate or GED	0.80	0.15, 4.11	0.794
Associates degree or 2-year certificate	0.51	0.06, 4.60	0.540
Some college	0.38	0.07, 1.76	0.230
College graduate	0.84	0.11, 6.48	0.861
Other	0.00	0.00, Inf	0.990
HEI Score			
<51: Low diet quality	—	—	
51–80: Moderate diet quality	0.42	0.11, 1.41	0.177
>80: High diet quality	0.05	0.01, 0.32	0.003
Total Comorbidities	1.51	1.16, 2.03	0.004
(**b**)			
**Characteristic**	**OR ^2^**	**95% CI ^2^**	***p*-Value ^3^**
Education			
Graduate or professional degree	—	—	
Eighth grade or less/some high school	1.06	0.23, 5.89	0.939
High school graduate or GED	1.29	0.47, 3.63	0.623
Associates degree or 2-year certificate	0.40	0.15, 1.01	0.055
Some college	0.90	0.36, 2.25	0.823
College graduate	1.90	0.70, 5.46	0.218
HEI Score			
<51: Low diet quality	—	—	
51–80: Moderate diet quality	2.62	0.99, 6.88	0.050
>80: High diet quality	2.25	0.63, 8.38	0.213
Missing	0.91	0.26, 3.11	0.879
Total Comorbidities	1.30	1.07, 1.60	0.012

(**a**) ^1^ Observations with missing DS data, BMI, and total comorbidities were excluded from model fitting (N = 13); ^2^ OR = odds ratio, CI = confidence interval; ^3^ A *p*-value < 0.05 suggests that there is sufficient evidence of greater/lower odds of the outcome happening at a 5% level of significance. (**b**) ^1^ Observations with missing DS data, BMI, and total comorbidities were excluded from model fitting (N = 31); ^2^ OR = odds ratio, CI = confidence interval; ^3^ A *p*-value < 0.05 suggests that there is sufficient evidence of greater/lower odds of the outcome happening at a 5% level of significance.

## Data Availability

This study is a secondary analysis of data collected in the Men Moving Forward and Moving Forward trials. The data presented in this study are not publicly available due to privacy and ethical restrictions related to participant confidentiality but may be obtained upon reasonable request and approval from the study’s principal investigators.

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
