# Peer review of "Exploring Dietary Supplement Utilization Patterns Among African American Survivors of Prostate and Breast Cancer: A Cross-Sectional Analysis"

_nutrients, 2025, doi:10.3390/nu17233724_

Round 1

Reviewer 1 Report

Comments and Suggestions for Authors

Manuscript ID: nutrients-3975109

It is a pleasure to review the article entitled “Exploring Dietary Supplement Utilization Patterns Among African American Survivors of Prostate and Breast Cancer: A Cross-Sectional Analysis.” This manuscript provides valuable insights into dietary supplement use among African American cancer survivors, a population that is often underrepresented in survivorship research. The topic is both timely and relevant, and the analytic approach is generally rigorous. In my view, these findings are important and may contribute to establishing guidelines for dietary supplement use among cancer survivors. However, several concerns need to be addressed, which are detailed below.

C1. In the abstract, please clarify how the denominator for supplement use prevalence was determined. The stated prevalence (e.g., 215 out of 376; which is 57.2%) does not align with the 63.8% figure in the results, which appears to exclude participants with missing data. Currently, the abstract and results use different denominators (376 vs 337), leading to confusion. Please consistently state and justify the denominator used for prevalence calculations and clearly note any exclusions due to missing data.

Material and Methods

C2. Please indicate the total number and percentage of participants excluded from analyses due to missing supplement data at the beginning of the section.

C3. The authors should specify in the statistical analysis subsection how missing data were handled for all major variables (e.g., “Participants with missing DS use data were excluded from prevalence and regression analyses.”). Consider adding a flow diagram.

C4. The authors should confirm that eligibility criteria match the original protocol (e.g., “at least six months prior to enrollment”) and correct any discrepancies.

Results

C5. The authors should use a consistent denominator (participants with available DS data) throughout the results when reporting prevalence rates. Explicitly state exclusions due to missing data in all tables, figures, and narrative text.

Example: “215 of 337 (63.8%) participants with available data reported supplement use; 39 (10.4%) were excluded due to missing data.

C6. As noted for line 195, remove “(N=376)” from the description of participants with reported DS use, as this number does not reflect only those with available data.

C7. In lines 223-224, consider revising the sentence to a more formal construction, such as “No statistically significant association was observed between education level and dietary supplement use among BC survivors.”

Discussion

This section could be strengthened by drawing more explicit comparisons between ‎the findings of this study and relevant previous research, highlighting both consistency and deviations from existing knowledge.‎ The authors should consider restructuring some of the longer and more complex sentences to enhance clarity and readability.‎

C8. Please discuss how gender may influence supplement use, as your comparison between breast and prostate cancer survivors (MF and MMF respectively) could be partly explained by gender differences.

C9. Consider rephrasing the sentence (lines 245-248) comparing dietary supplement use in your study and NHANES data for greater clarity.

C10. Consider revising sentences for improved clarity:

In lines 256–261, please revise the sentences to eliminate repetition and improve readability. The current text repeats terms and ideas about Non-Hispanic White samples and supplement use, which makes it difficult to follow.

In lines 292-295, please revise sentences for readability: “Despite high use of DS among cancer survivors, the existing evidence for the risks and health benefits of DS use for cancer survivors is often difficult to draw conclusions from. Some DS contain micronutrients found in foods, while others do not; some are developed from sources other than foods, and some contain other additional substances.”. Also provide specific examples of the other constituents found in dietary supplements, including appropriate references to support these sentences.

C11. In lines 299-301, please provide at least one reference to support the statement regarding Vitamin C and chemotherapy.

C12. Regarding line 313, consider revising for greater precision and clarity. For example, “A 2020 systematic review and meta-analysis of …”

C13. Please provide the full name of the abbreviation “PSA” used in line 325.

C14. Consider revising the phrase “preventing unnecessary costs and risks from DS use” to “preventing unnecessary costs and potential risks from DS use” to reflect the uncertainty.

C15. Explicitly discuss possible biases from missing DS data: acknowledge that excluded cases may differ from included participants, and consider the impact on prevalence estimates and generalizability of findings.

Reviewer 2 Report

Comments and Suggestions for Authors

In my opinion, it is a well-written manuscript, associated with robust statistical analysis, that explores an ongoing area of ​​research - cancer prognosis and survival in the context of dietary supplements as a potential intervention point in these patients.

To increase the quality of the manuscript, I have only a few minor suggestions for the authors:

- Please include age criteria for patient selection in the Study Design and Sample Population (section 2.1).

- Clearly define ages in years (in Table 1, the first row).

- Kindly explain why, sometimes, the percentage is not calculated in some cases in the same table (please refer to Marital Status or Godin Score)?

- Coloured Figures 1 and 2 will be more attractive to the readers.

- Please revise the reference list according to the recommendations of the Nutrients journal..

Reviewer 3 Report

Comments and Suggestions for Authors

This manuscript explores dietary supplement (DS) use among African American survivors of breast and prostate cancer. The topic is interesting, but the paper remains largely descriptive and lacks analytical depth.

  1. Clarify why eligibility criteria based on ACS guidelines and physical activity were applied to men only and not to women.
  2. The proportion of missing data on DS use (N=39) and HEI scores is unexpectedly high. Please explain why these key baseline variables were incomplete.
  3. The assessment of DS use requires more detail.
  4. The study is entirely descriptive and lacks a clear link to clinical or behavioural outcomes. The authors should clarify the broader purpose and contribution of documenting DS prevalence alone.
  5. The Discussion section extends into the risks and benefits of DS, which are not analysed in this study.
  6. The number of limitations and confounders is high; acknowledge this more explicitly in the Conclusions.
  7. Complete and alphabetize the list of abbreviations.
  8. Provide all cited supplementary materials (Tables S1–S3, Figures S1–S2).

The manuscript provides useful baseline data on DS use in an underrepresented population, but it remains too general and methodologically limited to fully inform clinical or public-health implications. Clearer methodological justification, improved handling of missing data, and more focused interpretation are needed.

Round 2

Reviewer 2 Report

Comments and Suggestions for Authors

The quality of the manuscript has greatly improved with more details and clarifications. I recommend the paper for publication in its present form. 

Reviewer 3 Report

Comments and Suggestions for Authors

I read your article and I found the topic potentially interesting and relevanti for the scientific community. However certain aspects require improvements before it can be considered for publication.

Introduction:

  • Lines 39-42 should be condensed as there are repetitions;
  • Lines 48-56 could be better summarized as there are a lot of repetitions;
  • DS section needs to be expanded;
  • At the end of the paragraph you shall end explaining more explicitly the objectives and purposes of your study.

Materials and Methods

  • In this section you shall explain better how patients were recruited;
  • You shall explain how you correlated MF and MMF since they are from two different decades;
  • Lines 146-148 need to be better written;
  • You should consider to explain better what is statistical significant and what is not.

Results

  • In paragraph 3.1 it is better to avoid to state all percentages that you already show in Table 1, you shall focus on showing the differences and compare the data you collected;
  • Lines 190-196; as stated before you shall interpret the differences and what is relevant rather than state all percentages as it is repetitive.
  • Also paragraph 3.4 needs to be focused on interpretation rather than data, already present in tables.

Discussion

  • In general the discussion is too long;
  • Lines 303-308 that repeat data are not necessary;
  • Lines 319-347 should not be part of the discussion; or need to be shortened. You shall state only the reasons that explain why it is difficult to draw conclusions regarding risks and benefits of DS;
  • You should try to connect better the paragraphs as this discussion section seems not to be easy to read;
  • Lines 380-383 are already in your conclusion part so can be removed. 
